

# Exogenous hormones influence *Brassica napus* leaf cuticular wax deposition and cuticle function

Zheng Yuan[1], Youwei Jiang[1], Yuhua Liu[1], Yi Xu[1], Shuai Li[1], Yanjun Guo[1], Reinhard Jetter[2,3] and Yu Ni[1]

[1] College of Agronomy and Biotechnology, Academy of Agricultural Sciences, Southwest University, Chongqing, China
[2] Department of Botany, University of British Columbia, Vancouver, Canada
[3] Department of Chemistry, University of British Columbia, Vancouver, Canada

## ABSTRACT

**Background**. Cuticular waxes cover plant surface and play important roles in protecting plants from abiotic and biotic stresses. The variations of wax deposition and chemical compositions under changing environments have been shown to be related to plant adaptations. However, it is still not clear whether the wax depositions could be adjusted to increase plant adaptations to stressed conditions.

**Methods**. In this study, exogenous methyl jasmonate (MeJA), the ethylene precursor 1-aminocyclopropane-1-carboxylic acid (ACC) and salicylic acid (SA) were applied to test their effects on cuticular wax deposition in two *Brassica napus* cultivars, Zhongshuang 9 (ZS9, low wax coverage) and Yuyou 19 (YY19, high wax coverage). Next, we measured the water loss rate and the transcriptional expression of genes involved in wax biosynthesis as well as genes related to disease defense.

**Results**. Seven wax compound classes, including fatty acids, aldehydes, alkanes, secondary alcohols, ketones, and unbranched as well as branched primary alcohols, were identified in *B. napus* leaf wax mixtures. MeJA, SA and ACC treatments had no significant effect on total wax amounts in YY19, whereas ACC reduced total wax amounts in ZS9. Overall, hormone treatments led to an increase in the amounts of aldehydes and ketones, and a decrease of secondary alcohol in ZS9, whereas they led to a decrease of alkane amounts and an increase of secondary alcohol amounts in YY19. Concomitantly, both cultivars also exhibited different changes in cuticle permeability, with leaf water loss rate per 15 min increased from 1.57% (averaged across treatments) at 1.57% (averaged across treatments) at 15 min to 3.12% at 30 min for ZS9 (except for ACC treated plant) and decreased for YY19. MeJA-treated plants of both cultivars relatively had higher water loss rate per 15 min when compared to other treatments.

**Conclusion**. Our findings that *B. napus* leaf wax composition and cuticle permeability are altered by exogenous SA, MeJA and ACC suggest that the hormone treatments affect wax composition, and that the changes in wax profiles would cause changes in cuticle permeability.

Corresponding author
Yu Ni, nmniyu@swu.edu.cn, nmniyu@126.com

## INTRODUCTION

*Brassica napus* L. is one of the most important oil crops grown worldwide for food, biofuels, lubricants and surfactants (*Allender & King, 2010*). However, abiotic stress from exposure to drought (*Shamloo-Dashtpagerdi, Razi & Ebrahimie, 2015*), heat (*Aksouh et al., 2001*), cold (*Scottl & Laimas, 2008*), and biotic stress from pathogen (*Bom & Boland, 2000*) and herbivore impact (*McInnes & Jamie, 2015*) severely reduce the yields of this crop. Plant species have evolved various physiological and biochemical mechanisms to avoid or tolerate the adverse effects of the above stresses (*Farooq et al., 2009*; *Hasanuzzaman, Hossain & Fujita, 2010*; *Kumar et al., 2017*). In many of these mechanisms, plant hormones, such as jasmonic acid (JA), ethylene (ET) and salicylic acid (SA), are known to regulate crucial elements of plant stress responses (*Creelman & Mullet, 1995*; *Kumar, 2013*). Furthermore, the intricate crosstalks among these hormones by which they can modulate growth and development in response to diverse environmental stresses have emerged as a common theme (*Jiang & Asami, 2018*). However, though these hormones regulate practically many aspects of plant stress responses, the effects of SA, JA and ET on the first protective barrier, plant cuticular wax, still remain unclear.

An important first line of plant defense against biotic and abiotic stress is the cuticle, a lipid coating of all primary, above-ground plant parts (*Shephered & Griffiths, 2006*; *Heredia & Dominguez, 2009*; *Reina-Pinto & Yephremov, 2009*). Plant cuticles consist of a scaffold of insoluble cutin, with cuticular waxes embedded and deposited on top (*Yeats & Rose, 2013*). The cuticular waxes are complex mixtures of very-long-chain aliphatics, with chain lengths ranging from $C_{20}$ to near $C_{40}$, comprising fatty acids and their derivatives such as aldehydes, alkanes, secondary alcohols, ketones, primary alcohols, and alkyl esters (*Jetter, Kunst & Samuels, 2006*). The relative amounts of the different compound classes and the chain length distributions within them differ drastically between plant species (*Jetter, Kunst & Samuels, 2006*). These aliphatic wax components are synthesized from $C_{16}$-$C_{18}$ fatty acids in the plastid, followed by fatty acid elongation in the endoplasmic reticulum to form VLCFAs. Then, the VLCFAs in the epidermal cells are converted to other wax products through decarbonylation pathway and acyl reduction pathway (Fig. 1) (*Millar et al., 1999*). In the decarbonylation pathway, VLCFAs are catalyzed into alkanes by a CER1, CER3 and Cytb5 complex (*Bernard et al., 2012*), and the alkanes are oxidized into secondary alcohols and ketones by a midchain alkane hydroxylase1 (MAH1) (*Greer et al., 2007*). In the acyl reduction pathway, VLCFA are catalyzed into primary alcohols and wax esters by CER4 and WSD1 (*Rowland et al., 2006*; *Li et al., 2008*).

It has long been established that the waxes form the major transport barrier within the cuticle and are thus the primary component for protection against biotic and abiotic stress (*Schönherr, 1976*). Various methods have been used to quantitatively measure the permeability of plant cuticle, relying on isolated cuticle membranes that can only be prepared from relatively sturdy leaves, or else involving radioactive tracer experiments (*Reed & Tukey, 1979*; *Stammitti, Garrec & Derridj, 1995*; *Riederer & Schreiber, 2001*). For phenotype characterization of wax-deficient mutants of *Arabidopsis thaliana*, more qualitative assays for cuticle permeability have been established, which gauge the staining
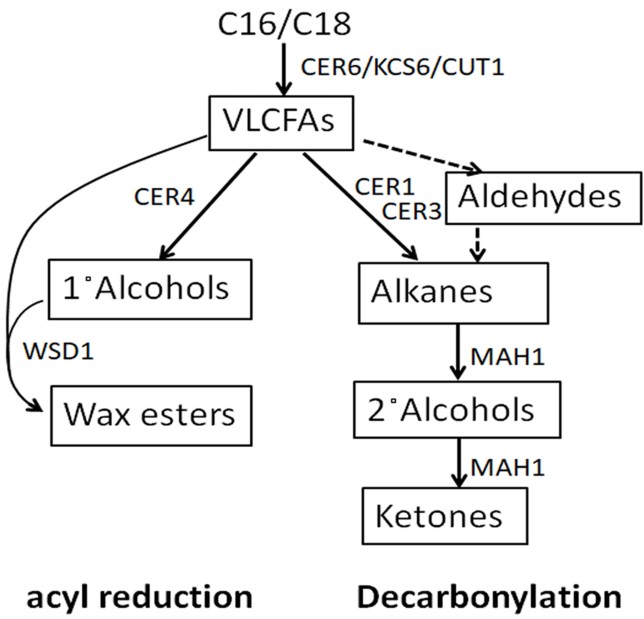

**Figure 1  Cuticular wax biosynthetic pathways.** Text in rectangular box denotes wax compound classes. Small text outside rectangular box denotes genes involved in wax production.

of the leaf tissue after surface application of the lipophilic dye toluidine blue (*Tanaka et al., 2004*), the rates of chlorophyll leaching into ethanol solution (*Lolle, Hsu & Pruitt, 1998*; *Aharoni et al., 2004*), or the rates of water loss from whole leaves or shoots (*Chen et al., 2003*; *Aharoni et al., 2004*). From the comparative analyses of diverse plant species and mutants with largely differing wax compositions, it could so far not be assessed how specific wax constituents and their relative amounts contribute to the eco-physiological functions of the cuticle.

There is also substantial debate around the question in how far the composition of cuticular waxes is dynamic, with possible acclimatization to adverse conditions either during organ growth or even afterwards. For example, the cuticular wax loads on *A. thaliana* leaves increased under drought, salt or osmotic stress conditions, or after exogenous application of abscisic acid (ABA) (*Kosma et al., 2009*), whereas decreased under dark conditions (*Go et al., 2014*). Accordingly, water deficiency was shown to induce the expression of cuticular wax biosynthesis genes (involving ABA), enhancing the accumulation of leaf cuticular waxes (*Seo et al., 2011*). A few earlier reports suggested that such dynamic responses of cuticular wax synthesis to adverse environmental conditions may also be mediated by other phytohormones. For example, *Cajustea et al. (2010)* reported that ET increased the total wax content in 'Navelate' orange fruit, along with structural changes in surface wax and likely effects on the physical barrier properties of the cuticle.

To clarify whether cuticular wax is also involved in the JA/ET or SA-dependent signaling pathways, we activated respective hormonal signaling pathways by exogenous application of SA, methyl jasmonate (MeJA), and the ET precursor 1-aminocyclopropane-1-carboxylic acid (ACC), before analyzing the cuticular wax composition and cuticle permeability of *B*.

*napus*, as well as the transcription expression of genes involved in cuticular wax biosynthesis. The main objective of this study was to analyze whether the alteration of cuticular wax induced by exogenous phytohormone would affect the functions of cuticle. Considering that plant genotypes play important roles in controlling cuticular wax deposition (*Andrae et al., 2019*; *O'Toole & Cruz, 1983*), two cultivars with different wax coverage were selected to test whether or not the responses of cuticular wax to these defense signaling molecules were related to plant genotypes. We hypothesized that exogenous SA-, JA-, and ET could adjust the cuticular wax depositions and thus increase plant adaptations to stressed conditions.

## MATERIALS AND METHODS

### Plant materials

Seeds of the *B. napus* cultivars Zhongshuang 9 (ZS9) and Yuyou 19 (YY19) were provided by the Chongqing Rapeseed Engineering Research Center, Beibei, Chongqing, China. Both of the cultivars were fall sowing cultivars in Southwest China with similar growing period from late September to early May next year. YY19 was yellow-seeded and ZS11 was black-seeded. The seeds were germinated on moist filter paper in Petri dishes for 7 d, and then healthy seedlings were transferred to pots with one plant in each pot. Each pot was filled with 0.2 kg soil (soil: peat = 2:1). The soils (Xanthic ferralsols, soil taxonomy) were sampled from fields growing corn, mixed with peat, then sterilized at 121 °C for 30 min. In total, there were 80 pots for each cultivar. The plants were then placed in a growth chamber at 22 °C in 16/8 h light/dark cycles for 35 d until the plants reached four-leaf stage. The plants were irrigated with tap water every five days, keeping the soil moisture content ca. 75% field capacity. The positions of the pots were changed every two days. No significant difference in plant heights and leaf numbers could be observed between two cultivars, except for their difference in leaf area.

### MeJA, SA and ACC treatments

The experiment was a two-way completely randomized design with two cultivars and four treatments, irrigating with 1% MS (Murashige and Skoog medium) solution (control), 0.1 mM MeJA (first dissolved in 0.1% ethanol, then diluted in 1% MS solution), 0.2 mM SA or 0.2 mM ACC (all from Sigma-Aldrich) solutions (in 1% MS solution) once every 3 d and in total 7 times. Each time, 20 ml solutions were irrigated directly into pot using cylinder, totally 140 ml for each pot. Adding hormones directly into soils has been shown to be effective in study of *Macková et al. (2013)*. The plants were at four-leaf stage when being subjected to hormone treatments. All plants grew well, to roughly equal heights and without showing nutrient deficiency symptoms. During the 21-day treatment period, the size of the second leaf (from the top) increased from $30.19 \pm 2.92$ cm$^2$ to $39.94 \pm 3.09$ cm$^2$ on average for ZS9 and from $31.92 \pm 1.45$ cm$^2$ to $56.68 \pm 1.07$ cm$^2$ for YY19. After 21 d of hormone treatment, the fourth leaf (from the top) of each plant (i.e., the second leaf from the top at the time of treatment) was harvested for cuticular wax analysis and cuticle permeability assessment. Two new leaves emerged during 21 days and the leaf area of the fourth leaf almost stopped increasing after 21 days. Previous study indicated that

cuticular wax deposition among leaves at different positions showed no obvious difference in amount or composition for *B. napus* (*Ni et al., 2014*).

## Wax extraction and chemical analysis

Four pots in each treatment were used for wax extraction. The fourth leaf was cut using scissor, washed under tap water, then photos of the leaf were taken for pixel counting using ImageJ (*Abramoff, Magelhaes & Ram, 2004*) to determine surface areas. After that, whole leaves were submerged in chloroform (6 mL, Sigma) containing 5 µg *n*-tetracosane (Sigma) as an internal standard for 30 s, and a second time for 30 s in fresh chloroform (6 mL). The two extracts were combined and filtered through glass wool, and the solvent was removed using a nitrogen stream. Next, mixtures were derivatized with 20 µL N,O–bis(trimethylsilyl) trifluoroacetamide (BSTFA) and 20 µL pyridine for 45 min at 70 °C, and then surplus reagents were evaporated under nitrogen. Finally, the reaction mixture was dissolved in 200 µL chloroform for qualitative and quantitative wax analysis.

For qualitative analysis, 2 µL of the derivatized wax mixture was injected on a GCMS-QP2010 Ultra (Shimadzu, Japan) using a HP-5MS capillary column with length 30 m, inner diameter 0.32 mm, and film thickness 0.25 µm (Agilent Technologies, USA) and Helium as carrier gas (1.0 mL/min; 65.2 kPa). The GC oven temperature was programmed with an original temperature of 80 °C, increased at 15 °C min$^{-1}$ to 260 °C, constant for 10 min, then increased at 2 °C min$^{-1}$ to 290 °C, further increased at 5 °C min$^{-1}$ to 320 °C, and finally constant for 10 min. A split/split less injector was used with split ratio 1:3 and temperature set at 300 °C. Compound identification was based on comparison of mass spectra with published data and authentic standards.

Quantitative analyses were carried out as above, but with a 9790 II gas chromatograph (Zhejiang Fuli Analytic Instruments, China) using a DM-5 capillary column with length 30 m, inner diameter 0.32 mm, and film thickness 0.25 µm (Dikma Technologies, USA). The flame ionization detector (FID) temperature was set at 320 °C. Total wax amounts were expressed as micrograms per total leaf area ($\mu g/cm^{-2}$), calculated as averages of four biological replicates with standard deviations.

## Measurement of water loss

To quantify leaf water loss, plants from another seven pots in each treatment for each cultivar were dark-acclimated for 3 h prior to measurement. Leaves of hormone-treated and control plants were soaked in water for 60 min in a dark growth chamber at 22 °C and 65% relative humidity, then gently blotted dry, weighed every 15 min for 150 min, finally dried at 70 °C overnight to constant weight (dry weight). Total water content was calculated as the difference between initial water-saturated weights minus dry weights. The water loss at each time point was expressed as the percentage of the water loss per 15min relative to the total water content (*Zeisler-Diehl, Migdal & Schreiber, 2017*), calculated as averages of six replicates with standard deviations.

## Quantitative real-time-polymerase chain reaction (qRT-PCR)

Total RNA was isolated from leaves using TransZol kit (TransGen, China) and then was treated with DNase I (Takara, China). Purified RNA was reverse transcribed in a

10 µl reaction using an oligo-dT18 primer and first-strand cDNA was synthesized using PrimerScript RT reagent Kit according to the manufacturer's instructions (Takara, China). Q-PCR amplifications were performed on a 96-well plate with a Bio-Rad CFX96 real-time PCR system. Each reaction contained 10 µL of SYBR Premix Ex Taq (Takara, China), 2 µL of cDNA samples (diluted to 5ng µL$^{-1}$) and 0.8 µL of gene-specific primers (10 µM) in a final volume of 20 µL. Gene-specific primers were designed according the conserve region of each gene family or subfamily (Table S1). The thermal profile consisted of one cycle at 95 ° C for 30 s followed by 40 cycles of 95 °C for 5 s, 57 °C for 30 s and 72 °C for 20 s. The reference gene actin7 was used to normalize for differences of the total RNA amount. The relative quantities of gene expression for sample comparison were calculated using the comparative Ct ($2^{-\Delta\Delta Ct}$) method (*Wong & Medrano, 2005*). In total, there were two technical replicates for each biological replicate with three biological replicates in each treatment.

## Statistical analysis

Two way ANOVA analysis was applied to analyze the effects of cultivar and hormone treatment on cuticular wax and water loss and their interactions. One way ANOVA analysis was further applied to compare the effects of hormone treatments on amounts of total wax and wax compositions according to the least significant difference test ($P < 0.05$) using SPSS software version 15.0 (SPSS Inc., Chicago, USA).

# RESULTS

## Effects of SA, MeJA and ACC on cuticular wax-associated gene expression in *B. napus*

RNA was extracted from the leaves of ZS9 and YY19 treated by exogenous application of SA, MeJA and ACC. Genes related to the wax biosynthesis pathway, homologs of *Arabidopsis CER1*, *CER3*, *MAH1*, *CER4*, and *CER6*, were checked for their transcription level in both *B. napus* cultivars.

On exposure to SA, MeJA, and ACC, the decarbonylation pathway gene *BnCER1-1/2* (KF724897/ KT795330), *BnCER3* (KT795332) and *BnMAH1-1/2* (KT795344/ KT795345) were down-regulated in ZS9, while up-regulated in YY19 (Fig. 2). The VLCFA elongase component encoding gene *BnCER6-1/2* (KT795339/ KT795340) and the acyl reduction pathway gene *BnCER4-1/2* (KT795333/KT795334) were found down-regulated by SA, MeJA, and ACC in both cultivars, except for insignificant changes of *BnCER6-1/2* in MeJA treated ZS9. SA-responsive *PR1*, JA-responsive *PDF1.2* and ET-responsive *ERF2*, were significantly up-regulated by SA, MeJA, and ACC, respectively, in both cultivars (Fig. S1).

## Amounts of total wax and wax compositions

Total wax amounts on hormone-treated and untreated leaves of *B. napus* cultivar YY19 ranged from 20.95 to 24.16 µg cm$^{-2}$, which were significantly higher than those of ZS9 (ranging from 13.83 to 17.84 µg cm$^{-2}$) (Table 1, Fig. 3). Significant interactions existed between cultivar and hormone treatments on total wax amounts. MeJA, SA and ACC treatments had no significant effect on total wax amounts in YY19, whereas ACC reduced total wax amounts in ZS9.

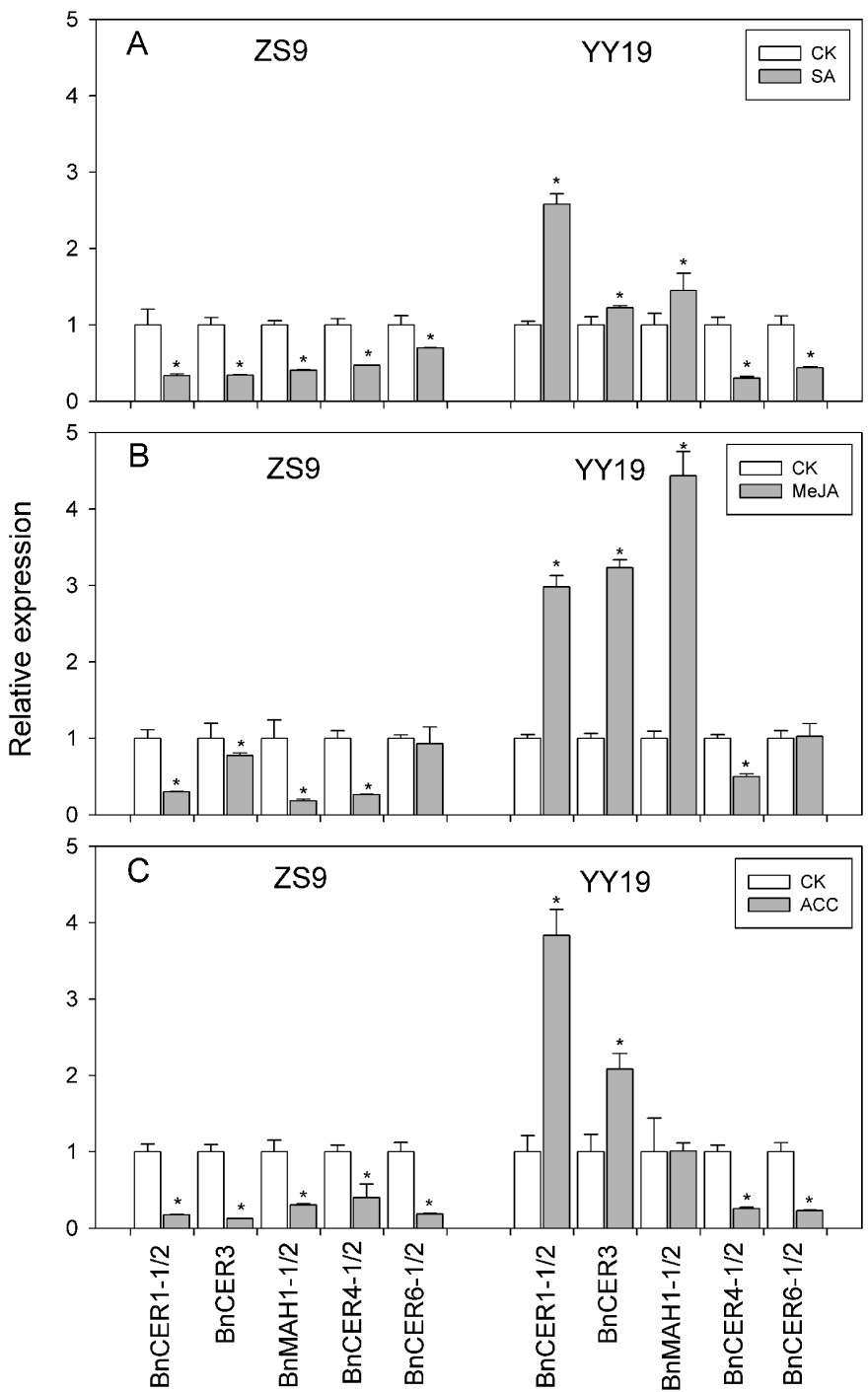

**Figure 2** **Effects of SA (A), MeJA (B) and ACC (C) on cuticular wax associated gene expression in Zhongshuang 9 (ZS9) and Yuyou 19 (YY19).** Plants were treated with 0.2 mM SA, 0.1 mM MeJA or 0.2 mM ACC for 21 d and plants treated with 1% MS were as control. The data represented the average of three biological replicates plus/minus standard deviation. Bars with asterisk represent significance at $P < 0.05$ according to student $T$ test when compared with the control for each gene.

**Table 1** ANOVA analysis of the effects of cultivar and hormone and their interactions on amounts of total wax, wax compositions, predominant wax compounds in each wax class, and leaf water loss ($F$ value).

| Parameter | Cultivar (C) | Hormone (H) | C × H |
|---|---|---|---|
| Total wax | 111.332*** | 2.129 | 4.062* |
| Fatty acids | 0.337 | 3.971* | 7.676*** |
| Aldehydes | 2.785 | 4.363* | 2.107 |
| Alkanes | 65.650*** | 7.192** | 4.467* |
| Secondary alcohols | 16.017*** | 4.699* | 14.927*** |
| Ketones | 2.863 | 1.934 | 0.304 |
| Primary alcohols | 54.654*** | 3.469* | 1.092 |
| 24-Methyl hexacosanol | 14.321** | 3.330* | 4.500* |
| Unidentified | 17.200*** | 1.049 | 8.491** |
| Nonacosane | 74.850*** | 3.630* | 5.391** |
| 1-Octacosanol | 80.741*** | 2.153 | 6.398** |
| 10-Nonacosanol | 12.665** | 4.570* | 13.604*** |
| Triacontanoic acid | 0.265 | 5.998** | 9.659*** |
| Triacontanal | 5.228* | 5.843** | 3.187* |
| Water loss | 0.032 | 7.548*** | 6.488** |

**Notes.**
Water loss used in the analysis were the data from the last measurements.

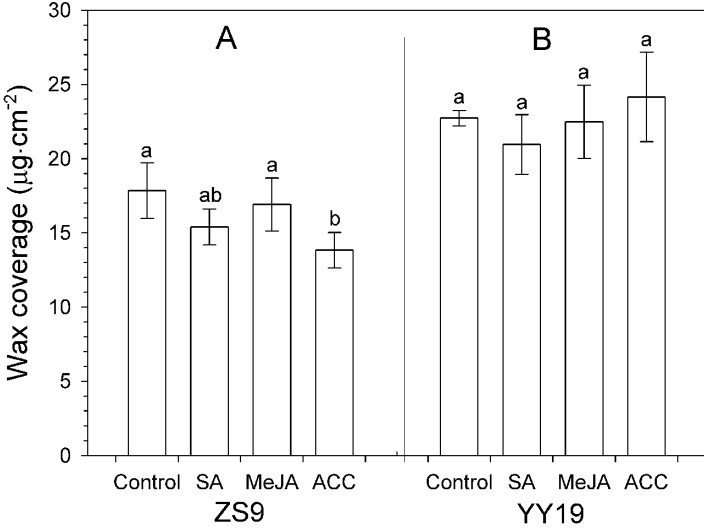

**Figure 3** **Total wax amounts on *Brassica napus* leaves treated with stress hormones.** (A) Cultivar Zhongshuang 9 (ZS9) and (B) Cultivar Yuyou 19 (YY19). Plants were treated with 0.2 mM SA, 0.1 mM MeJA or 0.2 mM ACC for 21 d, then leaves were extracted with chloroform and resulting wax solutions analyzed using GC-FID. The data represent the average of four biological replicates plus/minus standard deviation, and different lowercase letters above the data bars represent significance at $P < 0.05$ according to least significant difference test using one-way ANOVA analysis.

Seven classes of wax compounds were identified in the wax mixtures of all sampled *B. napus* leaves, including unbranched fatty acids, alkanes, secondary alcohols, ketones, aldehydes and primary alcohols, and branched primary alcohols. Both cultivars had similar wax compositions, with alkanes accounting for 55–69%. However, the amounts of wax compound classes shifted after MeJA, SA and ACC treatments (Fig. 4), and varied between cultivar and hormones (Table 2). For example, the amounts of aldehydes significantly increased in all hormone treated ZS9 plants, whereas unchanged in YY19 plants. The amounts of alkanes increased in MeJA-treated ZS9 plants, decreased in ACC-treated ZS9 plants and all hormone-treated YY19 plants, and unchanged in SA treated ZS9 plants. Concomitantly, the amounts of secondary alcohols was reduced after treatment in ZS9, and increased in YY19. The amount of ketone increased in MeJA- and SA-treated ZS9 plants, while they unchanged in hormone-treated YY19. The amounts of other compound classes did not change significantly upon treatment, excepting for a significant decrease of 24-methyl hexacosanol in hormone-treated ZS9 plants.

## Chain length profiles within wax compound classes

Detailed analyses of the chain length distributions within each compound class of the *B. napus* leaf wax revealed that the fatty acids and aldehydes were dominated by respective $C_{30}$ homologs, the alkanes, secondary alcohols and ketones by the $C_{29}$ homolog, the unbranched primary alcohols by the $C_{28}$ homolog, and the branched primary alcohols by the $C_{26}$ homolog (Table 2).

In the waxes of the hormone-treated plants, fatty acids with chain lengths ranging from $C_{26}$ to $C_{32}$ were detected (Table 2). The responses of fatty acid amount to hormones varied between hormones and cultivar. For example, the amounts of hexacosanic acid increased in hormone-treated ZS9 plants but unchanged in YY19, whereas the amounts of triacontanoic acid increased in ACC-treated ZS9, decreased in MeJA-treated YY19 plants, and unchanged for the other treatments. The amounts of octacosanal unchanged in MeJA-, SA- and ACC-treated ZS9 plants and reduced in MeJA-treated YY19 plants, the nonacosanal reduced in MeJA-, SA- and ACC-treated ZS9 plants but unchanged in YY19, while the amounts of triacontanal increased in MeJA-, SA- and ACC-treated ZS9 plants but unchanged in YY19. The alkane chain lengths ranged from $C_{27}$ to $C_{33}$, showing very strong odd-over-even predominance, with nonacosane and hentriacontane the two predominant alkanes. The amounts of nonacosane unchanged in hormone-treated ZS9 and YY19, whereas the amounts of hentriacontane increased in MeJA- and SA-treated ZS9 plants and reduced in MeJA-, SA- and ACC-treated YY9 plants. Secondary alcohols with chain lengths ranging from $C_{28}$ to $C_{31}$ were observed. The amounts of 10-octacosanol and 10-hentriacontanol unchanged in all hormone-treated ZS9 plants and YY19 plants (excepting for 10-hentriacontanol in ACC-treated YY19), while the amounts of 10-nonacosanol and 10-triacontanol reduced in hormone-treated ZS9 plants and increased in hormone-treated YY19 plants for 10- nonacosanol (insignificance for MeJA-treated YY19) and for 10-triacontanol in ACC-treated YY19 plants. The only ketone detected had chain length $C_{29}$ (nonacosanone) in both cultivars, the amount of which increased in SA- and MeJA-treated ZS9 plants and unchanged in YY19. Finally, $C_{26}$ to $C_{30}$ primary alcohols

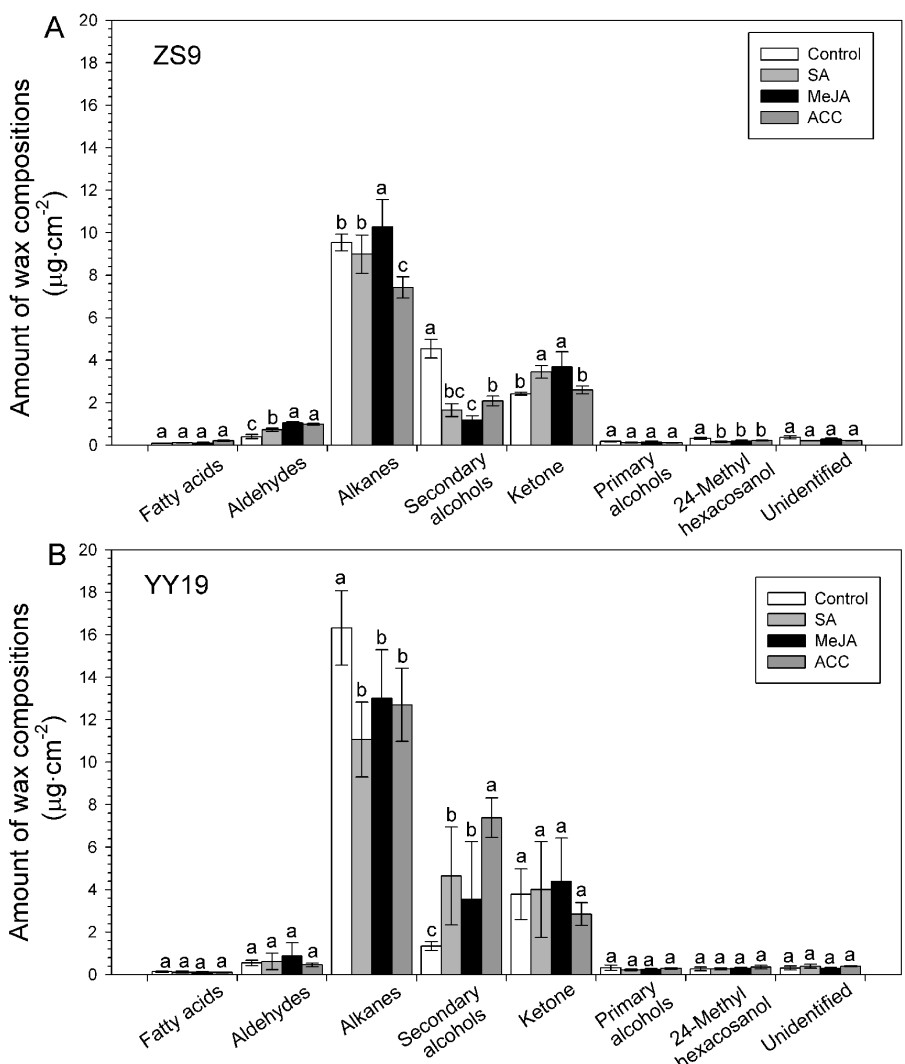

**Figure 4** **Amounts of wax compositions on *Brassica napus* leaves treated with stress hormones.** (A) Cultivar Zhongshuang 9 (ZS9) and (B) Cultivar Yuyou 19 (YY19). Plants were treated with 0.2 mM SA, 0.1 mM MeJA or 0.2 mM ACC for 21 d, then leaves were extracted with chloroform and resulting wax solutions analyzed using GC-FID. The data represent the average of four biological replicates plus/minus standard deviation, and different lowercase letters above the data bars represent significance at $P < 0.05$ according to least significant difference test using one-way ANOVA analysis.

were observed, the amounts of which showed no significant difference between the control plant and the hormone-treated plants in both cultivars, excepting for an increased trend of 1-hexacosanol in hormone-treated ZS9 plants.

## Effects of SA, MeJA and ACC on cuticle permeability

To assess the impacts of the stress hormones SA, MeJA and ET (or its proxy ACC) on *B. napus* cuticle permeability, we assayed leaf area, leaf weight, and leaf water loss rate. Leaf area and leaf weight were reduced in hormone-treated ZS9 plants, relative to the control, but remained unchanged in YY19 (Table 3). For ZS9, leaf water loss rate per 15

**Table 2** Amount of each wax compound on cultivar Zhongshuang 9 (ZS9) and Yuyou 19 (YY19) as affected by exogenous hormones ($\mu$g cm$^{-2}$).

| Compound | | Control | SA | MeJA | ACC |
|---|---|---|---|---|---|
| Hexacosanic acid | ZS9 | 0.012 ± 0.002c | 0.027 ± 0.004a | 0.023 ± 0.002a | 0.018 ± 0.002b |
| | YY19 | 0.016 ± 0.001ab | 0.020 ± 0.003a | 0.020 ± 0.005a | 0.012 ± 0.002b |
| Octocosoic acid | ZS9 | 0.030 ± 0.005b | 0.032 ± 0.004b | 0.032 ± 0.001b | 0.048 ± 0.007a |
| | YY19 | 0.027 ± 0.001a | 0.023 ± 0.004a | 0.028 ± 0.011a | 0.021 ± 0.006a |
| Triacontanoic acid | ZS9 | 0.053 ± 0.007b | 0.053 ± 0.017b | 0.068 ± 0.011b | 0.141 ± 0.028a |
| | YY19 | 0.116 ± 0.003a | 0.074 ± 0.050abc | 0.066 ± 0.017c | 0.077 ± 0.024ab |
| Lacceroic acid | ZS9 | 0.008 ± 0.002a | 0.011 ± 0.003a | 0.010 ± 0.002a | 0.010 ± 0.002a |
| | YY19 | 0.009 ± 0.001a | 0.009 ± 0.001a | 0.009 ± 0.001a | 0.006 ± 0.001b |
| Octacosanal | ZS9 | 0.088 ± 0.003a | 0.019 ± 0.004a | 0.019 ± 0.013a | 0.012 ± 0.006a |
| | YY19 | 0.080 ± 0.020a | 0.059 ± 0.027ab | 0.045 ± 0.014b | 0.069 ± 0.007ab |
| Nonacosanal | ZS9 | 0.038 ± 0.009a | 0.018 ± 0.004b | 0.023 ± 0.002b | 0.023 ± 0.001b |
| | YY19 | 0.047 ± 0.018ab | 0.044 ± 0.014ab | 0.038 ± 0.011b | 0.063 ± 0.011a |
| Triacontanal | ZS9 | 0.122 ± 0.039c | 0.650 ± 0.073b | 0.953 ± 0.063a | 0.898 ± 0.033a |
| | YY19 | 0.327 ± 0.200a | 0.435 ± 0.432a | 0.716 ± 0.662a | 0.202 ± 0.065a |
| Heptacosane | ZS9 | 0.058 ± 0.005a | 0.060 ± 0.010a | 0.052 ± 0.043a | 0.053 ± 0.005a |
| | YY19 | 0.122 ± 0.008b | 0.145 ± 0.009a | 0.125 ± 0.012b | 0.149 ± 0.014a |
| Octacosane | ZS9 | 0.049 ± 0.004a | 0.050 ± 0.010a | 0.053 ± 0.004a | 0.049 ± 0.003a |
| | YY19 | 0.063 ± 0.003a | 0.072 ± 0.007a | 0.078 ± 0.020a | 0.071 ± 0.011a |
| Nonacosane | ZS9 | 8.041 ± 0.416ab | 7.000 ± 0.754ab | 8.060 ± 0.1.220a | 5.612 ± 0.418b |
| | YY19 | 10.831 ± 0.535ab | 8.852 ± 0.1.428b | 10.897 ± 1.367ab | 11.605 ± 1.760a |
| Triacontane | ZS9 | 0.097 ± 0.015ab | 0.096 ± 0.007b | 0.116 ± 0.012ab | 0.146 ± 0.015a |
| | YY19 | 0.064 ± 0.039b | 0.123 ± 0.020a | 0.105 ± 0.006a | 0.118 ± 0.021a |
| Hentriacontane | ZS9 | 1.242 ± 0.099c | 1.732 ± 0.149ab | 1.954 ± 0.180a | 1.493 ± 0.087abc |
| | YY19 | 5.194 ± 2.164a | 1.834 ± 0.0.545b | 1.750 ± 0.965b | 0.711 ± 0.548b |
| Tritriacontane | ZS9 | 0.050 ± 0.006ab | 0.049 ± 0.007abc | 0.043 ± 0.013c | 0.074 ± 0.006a |
| | YY19 | 0.047 ± 0.012a | 0.043 ± 0.014a | 0.043 ± 0.006a | 0.050 ± 0.012a |
| 10-Octacosanol | ZS9 | 0.201 ± 0.016a | 0.163 ± 0.036a | 0.224 ± 0.070a | 0.150 ± 0.024a |
| | YY19 | 0.226 ± 0.122a | 0.268 ± 0.128a | 0.237 ± 0.088a | 0.232 ± 0.043a |
| 10- Nonacosanol | ZS9 | 4.088 ± 0.378a | 1.418 ± 0.324b | 0.891 ± 0.284b | 1.872 ± 0.248b |
| | YY19 | 0.878 ± 0.289c | 4.156 ± 2.328ab | 3.124 ± 2.697bc | 6.765 ± 0.900a |
| 10-Triacontanol | ZS9 | 0.244 ± 0.061a | 0.056 ± 0.009b | 0.071 ± 0.024b | 0.042 ± 0.003b |
| | YY19 | 0.235 ± 0.081b | 0.211 ± 0.095b | 0.188 ± 0.084b | 0.376 ± 0.091a |
| 10-Hentriacontanol | ZS9 | 0.006 ± 0.004a | 0.010 ± 0.003a | 0.004 ± 0.001a | 0.016 ± 0.004a |
| | YY19 | 0.003 ± 0.001b | 0.009 ± 0.007ab | 0.004 ± 0.001b | 0.014 ± 0.005a |
| Nonacosanone | ZS9 | 2.423 ± 0.069c | 3.445 ± 0.301ab | 3.685 ± 0.711a | 2.601 ± 0.188abc |
| | YY19 | 3.787 ± 1.187a | 4.004 ± 2.257a | 4.389 ± 2.040a | 2.853 ± 0.536a |
| 1-Hexacosanol | ZS9 | 0.005 ± 0.004b | 0.024 ± 0.005a | 0.014 ± 0.003ab | 0.016 ± 0.002ab |
| | YY19 | 0.076 ± 0.086a | 0.008 ± 0.007a | 0.017 ± 0.015a | 0.006 ± 0.003a |
| 1-Octacosanol | ZS9 | 0.151 ± 0.022a | 0.078 ± 0.024a | 0.114 ± 0.051a | 0.063 ± 0.009a |
| | YY19 | 0.196 ± 0.022ab | 0.185 ± 0.039b | 0.199 ± 0.049ab | 0.251 ± 0.028a |
| 1-Triacontanol | ZS9 | 0.021 ± 0.005ab | 0.014 ± 0.003b | 0.014 ± 0.003b | 0.028 ± 0.002a |
| | YY19 | 0.054 ± 0.028a | 0.030 ± 0.001a | 0.032 ± 0.011a | 0.038 ± 0.016a |

**Notes.**
The data represented the average of four biological replicates plus/minus standard deviation, and different lowercase letters after the values in each row represented significance at $P < 0.05$ according to least significant difference test.

**Table 3** Effects of hormone treatments on the surface area and water content of the fourth leaf of *B. napus*.

| Cultivar | Treatments | Leaf area (cm² per leaf) | Leaf weight (g per leaf) | Water content (%) |
|---|---|---|---|---|
| ZS9 | Control | 48.33 ± 2.51a | 1.90 ± 0.28a | 89.50 ± 0.62a |
| | SA | 40.82 ± 6.34b | 1.33 ± 0.17b | 89.56 ± 1.08a |
| | MeJA | 34.92 ± 9.17c | 1.49 ± 0.28b | 89.30 ± 0.62a |
| | ACC | 35.68 ± 3.50c | 1.35 ± 0.23b | 88.47 ± 1.13a |
| YY19 | Control | 58.13 ± 8.01a | 1.79 ± 0.14a | 91.06 ± 0.99a |
| | SA | 56.33 ± 4.32a | 1.74 ± 0.20a | 90.29 ± 0.98a |
| | MeJA | 53.81 ± 12.24a | 1.57 ± 0.36a | 91.73 ± 0.48a |
| | ACC | 58.48 ± 10.26a | 1.66 ± 0.19a | 91.04 ± 0.75a |

**Notes.**

The data represent the average plus/minus standard deviation. Different lower-case letters followed after values within each cultivar represent significance according to least significant difference test ($P < 0.05$).

min increased from 1.57% (averaged across treatments) at 15 min to 3.12% at 30 min for all plants except for ACC treated plants, reduced sharply at 45 min, and then followed by slight decrease thereafter (Fig. 5A). For YY19, leaf water loss rate per 15 min reduced slightly during 150 min (Fig. 5B). Overall, MeJA treated plants of both cultivars had higher water loss rate per 15 min when compared to other treatments.

## DISCUSSION

The expansion of aerial organs in plants is coupled with the synthesis and deposition of a hydrophobic cuticle, composed of cutin and waxes, which is critically important in limiting water loss (*Yeats & Rose, 2013*). In this study, the wax load ranged from 20.95 to 24.16 µg cm$^{-2}$ for YY19 and from 13.83 to 17.84 µg cm$^{-2}$ for ZS9. This was in consistent with the results of *Tassone et al. (2016)* where wax loads varied from 6.86 to 22.6 µg cm$^{-2}$ among 517 *B. napus* lines. Exogenous application of phytohormones, SA, MeJA and ethylene (or its proxy ACC), affected the wax compound class distributions within the wax mixtures and to some degree the chain length profiles within each compound class (Fig. 4, Table 2). It suggested that these hormones might be directly or indirectly involved in wax biosynthesis. The hormone treatments also led to changes in cuticle permeability. Higher cuticle permeability has been found to be associated with higher water loss (*Riederer & Schreiber, 2001*; *Weng et al., 2010*). The two *B. napus* cultivars tested in the current study responded differently to hormone treatments, particularly during the first 30 min, where ZS9 showed an increased permeability, whereas YY19 reduced. Concomitantly, most hormone treatments led to an increase in the amounts of aldehydes and ketones, and a decrease of secondary alcohol in ZS9, whereas led to a decrease of alkane amounts and an increase of secondary alcohol amounts in YY19. The changes in cuticle permeability in hormone-treated *B. napus* leaves were thus accompanied by shifts between wax constituents with varying polarities. A study on *Rosa canina* leaves indicated that the epicuticular wax contained higher concentrations of alkanes and alkyl esters but lower concentrations of primary alcohols and alkenols when compared to the intracuticular wax,

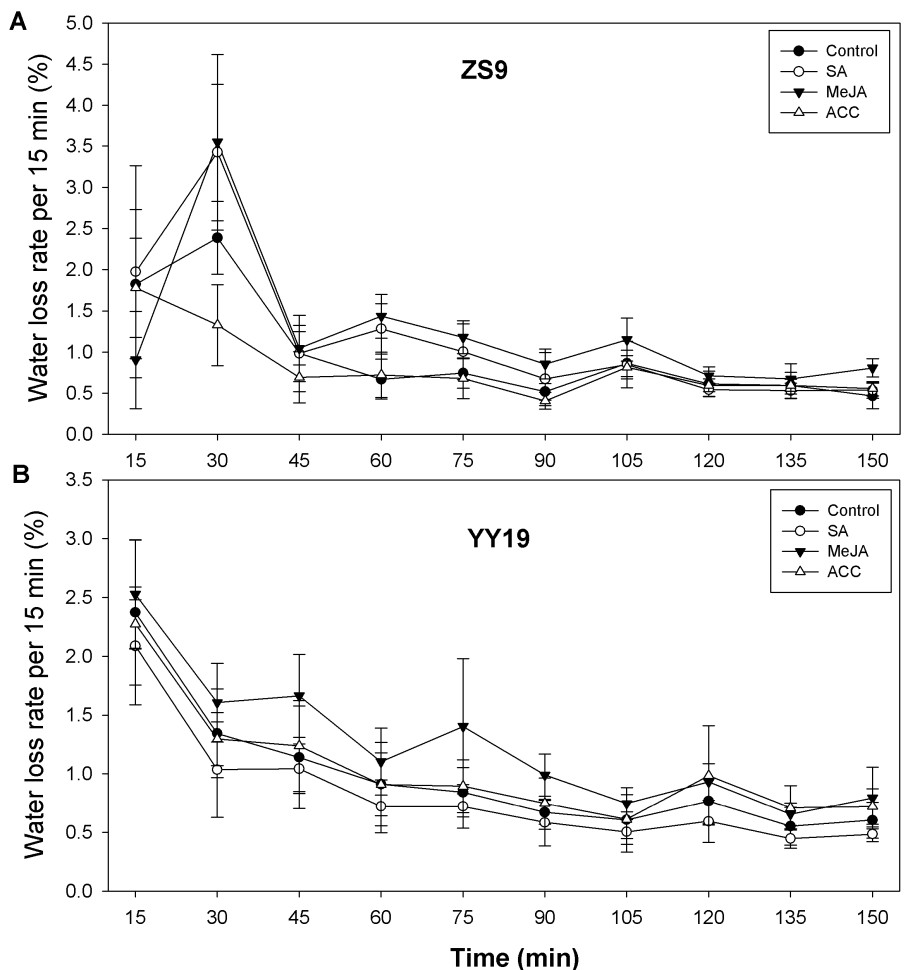

**Figure 5** **Water loss rates of isolated *Brassica napus* leaves treated with stress hormones.** (A) Cultivar Zhongshuang 9 (ZS9) and (B) Cultivar Yuyou 19 (YY19). Plants were treated with 0.2 mM SA, 0.1 mM MeJA or 0.2 mM ACC for 21 d, then water loss of isolated leaves was monitored every 15 min over 2.5 h. The water loss at each time point was expressed as the percentage of the water lost relative to the total water content. Data are shown as average of seven replicates plus/minus standard deviation.

resulting in the polarity differences (*Buschhaus, Herz & Jetter, 2007*). Different polarities of wax constituents contributed to the effectiveness of cuticular transpiration barrier, influencing the cuticle permeability (*Buschhaus & Jetter, 2012*; *Zeisler & Schreiber, 2016*). These findings suggested that the changes of cuticular wax constitutes induced by hormone treatments would alter cuticle barrier properties.

Relatively higher water loss rate per 15 min of plants treated with MeJA was observed for both cultivars, suggesting that a direct linkage between hormone treatment and the changes in wax compounds might exist. In fruits of *Capsicum* spp., *Parsons et al. (2013)* found differences in water loss among the accessions correlated with alkane amounts. Fruits of the tomato *ps* mutant had severely reduced amounts of *n*-alkanes (and aldehydes) together with a five- to eight fold increase in water loss per unit time and surface area

relative to wild type (*Leide et al., 2011*). Similarly, the alkane contents were dramatically reduced in the leaves of drought-susceptible *Arabidopsis* mutant *myb96-1*, while the corresponding *myb96-1D* gain-of-function line had elevated alkane quantity and was more drought-resistant than wild type (*Seo et al., 2011*). Conversely, it was also found that water deficit, sodium chloride, or ABA treatments led to significant increases in alkane amounts of the *Arabidopsis* wild type leaf wax (*Kosma et al., 2009*). Finally, *CER1*-overexpressing *Arabidopsis* plants showed drastically increased alkane amounts and reduced cuticle permeability together with increased susceptibility to bacterial and fungal pathogens (*Bourdenx et al., 2011*). However, an increase of alkanes in MeJA-treated ZS9 and a decrease of alkane in YY19 did not match with their changes of cuticle permeability. This might be attributed to the complicated crosslinking between hormones that are involved in regulating wax biosynthesis, which might offset the effects of one hormone by other hormones (*Wang & Irving, 2011*). On the other hand, the amount of alkane might be not always correlated with water loss. The physical properties of cuticle (such as thickness) or cutin amount might also be altered by hormones, which thus contributed to the cuticle barrier function (*Yeats & Rose, 2013*). Further studies are needed in the future to measure both the chemical profiles of cuticular wax and cutin as well as physical properties of the cuticle to analyze the influence of exogenous hormones on cuticle permeability. Though there is evidence from diverse species that the alkane amount within wax mixtures is positively correlated with cuticle permeability (*Leide et al., 2011*; *Parsons et al., 2013*), our results on *B. napus* leaves partly supported this conclusion but further implied that factors other than alkane might also contribute to cuticle permeability.

The hormone-prompted shifts in wax compound classes on *B. napus* leaves were accompanied by moderate changes in chain length profiles within the fraction (Table 2). However, different chain length compounds in each wax class responded differently to hormone treatments, leaving the average chain length across all compound classes largely unchanged for both cultivars (Table S2). Based on previous models (*Schreiber & Riederer, 1996a*; *Schreiber & Riederer, 1996b*), this narrower chain length distribution should lead to improved packing of molecules in crystalline wax domains and thus higher overall crystallinity, which in turn would be expected to result in lower permeability. However, these predictions were not confirmed by our findings that hormone-treated plants had increased permeability relative to the control together with narrower chain length profiles. We therefore conclude that the chain length differences observed here affected permeability much less than the shifts in compound class distribution (towards alkanes, see above).

It is believed that *CER1*, *CER3* and *MAH1* were responsible for the synthesis of alkanes, secondary alcohols and ketones (*Bernard et al., 2012*; *Greer et al., 2007*). The current study showed that an increase and a decrease in the levels of secondary alcohols were correlated with the enhanced and reduced expression of *BnMAH1-1/2* in decarbonylation pathway (Fig. 2). Meanwhile, the different responses of secondary alcohol level and ketone level under hormone treatments between the two cultivars also suggested that the efficiency of MAH1 in converting alkane to secondary alcohol and from secondary alcohol to ketone were different (*Greer et al., 2007*). The expression of *BnCER1-1/BnCER1-2* was not consistent with the changes of alkanes. This, partly, could be explained by the functional

divergence of other *CER1* homologues in *B.napus* and posttranslational modification. *Bourdenx et al. (2011)* reported that *CER1*-overexpressing plants showed reduced cuticle permeability together with reduced susceptibility to soil water deficit and increased susceptibility to bacterial and fungal pathogens. *Asselbergh et al. (2007)* suggested that a more permeable cuticle in an abscisic acid-deficient sitiens tomato mutant facilitated the release of an antifungal compound with an efficient fungistatic effect. In the current study, the SA-responsive *PR1* gene exhibited significant increase in transcript abundance in ZS9 and YY19 (Fig. S1). It seems that both the alteration of cuticular waxes and the induction of the antimicrobial compounds such as PR proteins are involved in the SA-mediated defense response.

It should be noted that the chemical and physiological differences between hormone-treated and control plants were not merely due to more general changes the hormones might have caused on growth of the plants (*Santner, Calderon-Villalobos & Estelle, 2009*; *Oliva, Farcot & Vernoux, 2013*). We found that the treated plants grew to the same height, and that the sampled leaves of treatment and control plants had similar relative water content. In the cultivar ZS9, treatment with all three hormones led to somewhat smaller leaf sizes (and consequently total leaf weights), which were, however, not reflected in total wax coverage. It thus appears that the decrease in leaf size is accompanied by a proportional decrease of the absolute wax quantity (per leaf), resulting in unchanged wax amounts per unit surface area. In the cultivar YY19, neither the leaf size nor the wax coverage was affected by hormone treatments. Overall, the chemical and physiological effects of hormone treatments could, thus, not be attributed to sample size effects due to differential plant growth after hormone treatment. Meanwhile, the hormones were added through the soils (*Macková et al., 2013*), instead of leaf spraying. Therefore, the changes of leaf cuticular wax on hormone-treated plants might be directly induced by the changes of endogenous hormones. Studies have shown that hormones like ABA and ETH are involved in cuticular wax biosynthesis (*Cajustea et al., 2010*; *Martin et al., 2017*). The MeJA, SA and ACC used in the current study might regulate the cuticular wax deposition by adjusting the levels of ABA or ETH through crosstalk among hormones (*Depuydt & Hardtke, 2011*).

## CONCLUSIONS

All taken together, our finding that exogenous hormones influence the *B. napus* cuticular waxes may suggest that the changes in cuticle composition and permeability were all due to a fundamental stress response of the growing epidermal cells. It may, therefore, be speculated whether the resulting changes in the wax composition and barrier properties were caused indirectly, and that both were parallel effects of a basal change rather than linked causally. However, based on the additional finding that the two cultivars showed different effects of both cuticle composition and permeability, it seems likely that the two effects are linked directly. We therefore conclude that the hormone treatments (particularly MeJA) affect wax composition, and that the changes in wax profiles cause changes in cuticle permeability.

**Abbreviations**

| | |
|---|---|
| **ABA** | abscisic acid |
| **ACC** | 1-aminocyclopropane-1-carboxylic acid |
| **BSTFA** | bis-N,O-trimethylsilyltrifluoroacetamide |
| **ET** | ethylene |
| **FID** | flame ionization detector |
| **JA** | jasmonic acid |
| **MeJA** | methyl jasmonate |
| **SA** | salicylic acid |

### Funding
This research was funded by the National Natural Science Foundation of China (31771694, 31670407) and the Chongqing basic and advanced research project (cstc2018jcyjA0857). The funders had no role in study design, data collection and analysis, decision to publish, or preparation of the manuscript.

### Grant Disclosures
The following grant information was disclosed by the authors:
National Natural Science Foundation of China: 31771694, 31670407.
Chongqing basic and advanced research project: cstc2018jcyjA0857.

### Competing Interests
The authors declare there are no competing interests.

### Author Contributions
- Zheng Yuan performed the experiments, analyzed the data, prepared figures and/or tables, and approved the final draft.
- Youwei Jiang, Yuhua Liu, Yi Xu and Shuai Li performed the experiments, prepared figures and/or tables, and approved the final draft.
- Yanjun Guo and Yu Ni conceived and designed the experiments, analyzed the data, prepared figures and/or tables, authored or reviewed drafts of the paper, and approved the final draft.
- Reinhard Jetter analyzed the data, authored or reviewed drafts of the paper, and approved the final draft.

### Data Availability
The raw measurements are available in the Supplemental Files. The raw data include cuticular wax on canola leaf, water loss rate, transcription of wax genes and hormone-responsive genes.

## Supplemental Information

Supplemental information for this article can be found online at http://dx.doi.org/10.7717/peerj.9264#supplemental-information.

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
