# Peer review of "Exogenous hormones influence Brassica napus leaf cuticular wax deposition and cuticle function"

_PeerJ, doi:10.7717/peerj.9264_

## Round 0.1 · original submission · Major Revisions

You got three very good and constructive reviews back. Please follow their advise when you revise your manuscript. Please add a pathway figure to the introduction and consider carefully the comments on the permeability experiments.

·

Basic reporting

An interesting and well written article, although the English could use a little polishing. I enjoyed reading the manuscript.

The introduction gives appropriate background on what is known concerning the role of hormones and environment on wax biosynthesis, and outlines the hypothesis being tested.
• The pathways of wax biosynthesis are not described. This would make it difficult for someone not familiar with wax biosynthesis to interpret the data, and to understand the precursor/product relationships between the various wax components. A simple figure in the text or as a Supplementary figure should be provided. This should include the gene targets for the qRT-PCR.

Experimental design

Materials, methods and experimental design.

The experimental design and technical aspects of the manuscript are well described and appropriate. It is good to see that the authors used GC rather than GC-MS for the quantitate analysis and set the plants up in a randomized design. The authors were careful to use a leaf that was still expanding, with active wax biosynthesis, for the work. Classical permeability assays were used, water loss, chlorophyll leaching and toluidine blue staining.
• The spot application of toluidine blue, rather than leaf or leaf segment immersion makes this assay rather subjective. It is not easy to see any difference in permeability based on the leaf pictures in figure 5. This is a common problem with this test, but the authors also provide data on water loss and Chlorophyll leaching. Are there any better pictures that could be used?
• In this work, the hormones were applied to the soil. It would be helpful if the authors could provide some evidence that is an acceptable application method and that plants can take up MeJA, SA and ACC when applied in this manner.
• For the qRT-PCR section, better identification of the target genes is needed. This is important because some targets appear to be members of gene families and because B. napus is a tetraploid. As an example, BnCER6 would correspond to BnaA07g24600D and BnaC06g25830D. Using EnsemblPlants would be one way to obtain this information, although the sequenced genotype is not exactly the same as the ones used in the paper. Alternatively, the authors could reference the target by its accession number.
• From the primer sequences provided in Table S1 it is clear that the primers may not be appropriate for the study. Primer targets were checked using both the NCBI BLAST algorithm (https://blast.ncbi.nlm.nih.gov) and BLAST at EnsemblePlants (https://plants.ensembl.org). The BnCER1 primer will amplify a single gene product, when two would be expected. A similar situation exists for the BnCER4 primer pair. It is highly recommended that the authors redesign the qRT-PCR primers to be gene specific, or to target both (or more) of the targets in this tetraploid species. Alternatively, evidence should be provided that the qRT-PCR is indeed targetting all of the relevant genes. If this cannot be provided, then the qRT-PCR should then be repeated to gain more accurate results. Use of ACTIN-7 as a control could also be reconsidered (see Yang et al 2014, Gene 538:113-122)

Validity of the findings

• The first question to ask is whether the soil based application of hormones was effective. The authors have addressed this by looking at expression of PR1, PDF1.2 and ERF2. This should be moved to the beginning of the result section rather than the end.
• It would be helpful if the authors could comment on the typical variation in wax amount seen in cultivars of B. napus. Are the wax loads of ZS9 and YY19 within the limits of typical variation, or are they unusual? Tassone at al 2016 (Industrial Crops and Products 79:77-83) would be useful to address this question.
The data presented in the manuscript indicates that there are some differences in wax load and composition in plants subjected to the hormone treatment compared to their controls. The two cultivars clearly respond differently.
• Differences in the expression levels of genes associated with wax biosynthesis were also reported, although this may need to be reconsidered (see above). It is difficult to interpret the data without confidence in the qRT-PCR results.
• The paper does not report the alkyl-esters (wax-esters) present in the cuticular wax. These can account for a significant proportion of the wax and play a role in permeability (Tassone et al 2016).
The authors studied cuticle permeability using a number of methods. This is good, but raises questions.
Please comment on:
• Water loss over 150 minutes is greater for control plants of ZS9 (11.5%) than control plants of YY19 (8%) (figure 3), whereas chlorophyll leaching over 150 minutes is slightly greater for YY19 than ZS9.
• The lines differ significantly in leaf area for the control plants (table 3) suggesting different growth characteristics.

Conclusion
• The authors comment that all three hormones have fairly similar effects on the cuticle. This is a little misleading as the individual hormones appear have different effects on wax composition (Figure 2B) and gene expression (Figure 6).
• The authors conclude that changes in wax profile caused changes in cuticle permeability (line 378). This is not supported by the data. Correlating cuticle permeability to wax chemical composition is speculative without considering cuticle thickness and structure. This should be assessed using electron microscopy.
• The evidence does supports the hypothesis that hormone treatment may influence wax composition, but in contrasting ways for the 2 varieties. This is an intriguing observation.

Additional comments

The authors describe selecting two lines with different wax coverage for study (line 106). These lines also seem to differ in leaf area (table 3) additional phenotypes should be reported if known. Ideally, near isogenic lines differing in wax load would be better for comparison.

Based on figures 1 and 2, hormone treatment of YY19 does not change wax load, but increases secondary alcohol levels with a corresponding decrease in total alkanes (primarily the C31 alkane). The lack of increase of the C29-ketone is worthy of comment considering MAH1 is thought to be responsible for the synthesis of both the secondary alcohol and ketone (Greer et al 2007)

Reviewer 2 ·

Basic reporting

The manuscript "Exogenous hormones influenced the cuticular wax depositions on Brassica napus leaf and the cuticle function" by Yuan et al. describes the characterization of two cultivars of Brassica napus and how they respond to different phytohormones in terms of leaf wax accumulation, leaf wax composition, epidermal permeability, and cuticle-related gene expression.

I appreciate that the language was clear and easy to understand. Overall, the English was quite good. A very minor review of the English should be sufficient to make it even more comprehensible.

The article is well structured with a logical flow. The majority of the figures are well designed and easy to understand. In figure 5 I found the close up images of toluidine blue staining (bottom rows of A and B) difficult to discern, the images were very dark and it was difficult to make out any of the detail (especially in printed form). I also noticed some extraneous lines in Figure 1. Apart from that, the figures were well prepared.

One detail that I think should be included in the figure legends is the statistics used for each analysis. These are described in the materials and methods but I find it useful to include details about whether a 1-way or 2 way ANOVA was used including the post-hoc analysis.

Experimental design

I do believe that this manuscript fits within the aims and scope of this journal.

I do, however, have several concerns.

1. The gap of knowledge that this research fills in is not clearly stated.
2. Some methods were not clearly defined
2.a. Lines 125 and 126: It was not clear what the phytohormones were dissolved in. MeJA is difficult to dissolve in water. Was MeJA perhaps dissolved in ethanol? Were the control plants watered with nutrients containing an equivalent amount of solvent (e.g. methanol)? (were these "vehicle controls" or "mock treatments"?
2.b. For the experiment on the effect of SA, MeJA, and ACC on cuticle permeability it is stated in the text (line 269) "....when compared with water sprayed leaves..." this seems to imply that for this experiment the phytohoromones were applied via spraying. However, there is no mention of a spray phytohormone application in the materials and methods, only an irrigation-based treatment. Can the authors please clarify?
3. I noticed in the raw data under the "water loss" tab that for the ZS9 cultivar replicate 3 of the CK (control) and replicate 7 of the MeJA treatment are the averages of the other 6 replicates. Can the authors please explain this?

Validity of the findings

I do feel that this research does present some interesting results. However, I feel like the presented data and conclusions aren't entirely in agreement and these discrepancies weren't well discussed in the paper.

One of the major assertions of the paper is that the data presented in the manuscript are in agreement with data from other species. Line 310-311: "our results on alkane contribution to cuticle barrier function match previous results from other species...." The general trend from other species is that there is a negative relationship between alkane content and epidermal/cuticular permeability. That is to say that plants with elevated alkane contents have less permeable epidermis/cuticles and vice versa. The data in this manuscript show that ACC treatment, of both cultivars, results in decreased amounts of alkanes yet the ACC-treated ZS9 cultivar demonstrates lower water loss and chlorophyll leaching rates than control plants (and plants from all other treatments). The authors conclude that their data are supportive of the model that alkane content is negatively correlated with cuticle permeability...the results do not support this conclusion.

I think that these results present a unique discovery but the discovery hasn't really been discussed or framed properly.

It is curious that the cultivars have different responses to phytohormones in terms of epidermal permeability and associated phenotypes. Incorporation of more discussion on this topic (i.e. why do the cultivars differ in these responses) would most certainly strengthen the manuscript.

Additional comments

Overall, I find that this manuscript presents some interesting results. However, I feel that addressing some of my concerns would certainly improve the quality of the manuscript.

Reviewer 3 ·

Basic reporting

No comment

Experimental design

No comment

Validity of the findings

Data are over-interpreted.

Additional comments

The manuscript by Yuan et al., presents a large set of analytical, physiological and molecular data to describe the effects of the plant “stress” hormones salicylic acid (SA), methyl jasmonate and ACC, an precursor of ethylen, on cuticular waxes and cuticle barrier properties. Using sophisticated GC-MS analysis the authors detected diverse changes in the monomeric composition of leaf waxes upon hormone treatments in two Brassica napus cultivars (ZS9 and YY19). The studies were complemented with physiological assays to evaluate potential changes in leaf cuticle permeability and molecular studies monitoring the expression of wax biosynthetic enzymes.

I enjoyed seeing the comprehensive analytical data set on the leaf wax composition. The data are appealingly presented in adequate graphs. I liked that appropriate statistical tests were applied but I wish standard deviations (biological variation) instead of standard errors would be shown in the future. I also have some concerns regarding the interpretation of the data and the conclusions drawn.

The data of the permeability assays presented in figure 3 and 4 look to me like two-phase-kinetics. According to Zeisler-Diehl et al. 2017 (https://doi.org/10.1093/jxb/erx282) the final water loss or Chlorophyll leakage values are less indicative, but the slope of the changes in the linear phase best describe the permeability. Using the slope in the linear phase as a measure of permeability a different picture arises with some hormone treatments increasing water permeability (e.g. MeJA) and other hormones slightly decreasing water permeability. These data should be re-evaluated.
Consequently some parts of the discussion may need to be edited. Furthermore based on the current data the general statement (line 301) that the changes in permeability (Line 295, decrease in ZS9) are accompanied by shifts between more and less polar monomers is not justified since the alkanes in ZS9 do not change.
As the effects of the hormone treatments on wax monomers are very different in the two cultivars a direct linkage between hormone treatment and changes in wax monomers is not established. The authors should consider investigating a dose depended response of the different hormones to support their interpretation.
Although exciting, the discussion of the expression data (Line 342ff) is a bit biased. The explanation of the discrepancy between high CER1, CER3 expression and a reduction in alkanes by the specificity of these enzymes for C29 compounds seems over-interpreted. First the authors used “primers designed to the conserved region of the gene family” and thus don’t know which specific CER1/CER1-like etc. they amplified in the expression studies. Secondly, in YY19 the nonacosane only slightly changes (decreases and increases) by the different hormone treatments (Table 2) but all three hormones drastically increase the expression level of CER1. In ZS9 MeJA strongly reduces the expression of CER1 but nonacosane remains unaffected.
In my opinion the conclusions (Line 370ff) do not represent the data. “all three hormones had fairly similar effects” is totally different from the data presented when looking at waxes, permeability and expression. This should be rephrased much more carefully.
Although the authors discussed potential effects of the hormones on leaf growth and development they should also consider hormone induced effects on the extractability or “metabolism” of compounds (crosslinking, ester formation/release, etc.) in their discussion.

Minor:
Line 29 write “ethylene” in abstract.
Line 77 permeability of plant cuticle rather than “surface waxes”
Line 129 “elicitor” is not correct in this context.
Line 168 what was the room humidity / how was it kept constant?
Line 310 rephrase. The contribution of alkanes was not specifically investigated.
Figure 6: Clarify “control”. Control conditions or control gene ACTIN7?

---

## Round 0.2 · Minor Revisions

Please revise especially the points raised for #4 and 7.

·

Basic reporting

No further comment

Experimental design

Generally good.

Validity of the findings

See general comments

Additional comments

The authors have thoroughly assessed and responded to the comments from the first review.

Addition of the pathway figure and the brief description of wax biosynthesis will certainly be helpful.

Point 4. Changing the contrast on the Toluidine blue figure has not really resolved the issue. I find the data unconvincing. Leaves have spots of different intensities and the spots shown in the magnified images could easily have been chosen to fit a hypothesis. This is really a problem of the technique and I have no criticism of the authors. I would suggest removal of this figure. Water loss and chlorophyll leaching assays appear more convincing,

Point 7. Adding accession numbers has improved the manuscript as it enables better evaluation of the target genes. Confidence in the choice of primers and target genes really requires an improved assembly of the B. napus genome sequence, which would be beyond the scope of this paper.

Minor improvement of the English is required.

---

## Round 0.3 · Major Revisions

I still recommend to please add a pathway figure to the introduction. In addition reviewer #3 is still not fully convinced about your response on comment 6 dealing with the permeability experiments. Please address the concerns dealing with the rate and whether data have been mixed up.

Reviewer 3 ·

Basic reporting

no comment

Experimental design

no comment

Validity of the findings

no comment

Additional comments

The resubmitted manuscript by Yuan et al., shows advantageous improvements. The reorganisation (e.g. expression first) make sense. Similarly the removal of TBO staining. Also most of my comments and of other reviewers have been addressed in a satisfactory way. This includes all minor editing, adding details and rephrasing and more carefully presenting statements.
However I am not completely satisfied by the response to comment 6) – permeability. In the response the authors talk about a water loss and chlorophyll-leaching RATE. This is exactly what I requested. A RATE is a value (e.g. % water loss) per time unit. Such calculated rates (e.g. % water loss per min or % chlorophyll leaked per hour) are not presented or listed in the manuscript. Figure 3A shows the accumulated % water loss over a selected time period, not water loss per time unit. E.g. After about 120 min the % water loss per 15 min might be very similar for all four treatments.
In addition the permeabilities shown for ZS9 are clearly two phases (e.g. positive Y-axis intercept of trend lines in Fig 3) and are not one “linear phase” as the authors comment. Obliviously the water losses of ZS9 are different in the first 30 min compared to the rest of the time. To make my point a bit more obvious to the authors I made a graph using their data (provided in file peerj-44364-total-data-hormone-canola_wax.xlsx, data sheet “water loss”) but starting at the 30 min time point/ using the 30 min time point as reference. From 30 min on the curve looks somehow linear. As a result ACC is almost identical to control whereas SA and MeJa cause higher water losses. Until here I only focused on Fig 3A but similar conditions and changes may apply for other figures.
However, I do not insist on my point of view and I might be totally misguided. Nevertheless the authors are free take this example to reconsider some views.

I might be also mistaken in reading the data provided in file peerj-44364-total-data-hormone-canola_wax.xlsx (data sheet “water loss”). Nevertheless I got the impression that either data or labels in that table are not consistent with data presented in the Figure 3A, or data sets are mixed up. Reading the numbers from the figure (e.g. using a ruler) and comparing them to the data in the excel sheet it looked to me that: What is “Control” in the data table (mean 10.4) is “ACC” in Fig 3A, “SA” in table is “Control” in figure, “ACC” in table (mean 8.4) is “MeJa” in figure and “MeJa” in table is SA” in figure. If this really happened, than data between varieties and treatments may better fit at the end.
However, I may have read that table in an incorrect way and everything is fine. If so I greatly apologize for the confusion.

Annotated reviews are not available for download in order to protect the identity of reviewers who chose to remain anonymous.

---

## Round 0.4 · accepted · Accept

Thank you very much again for your careful revision.